# Efficient Memristive Circuit Design of Neural Network-Based Associative Memory for Pavlovian Conditional Reflex

**DOI:** 10.3390/mi13101744

**Published:** 2022-10-15

**Authors:** Samiur Rahman Khan, AlaaDdin Al-Shidaifat, Hanjung Song

**Affiliations:** Department of Nanoscience and Engineering, Centre for Nano Manufacturing, Inje University, Gimhae 50834, Korea

**Keywords:** associative memory, Pavlovian conditional reflex, memristor, neural network

## Abstract

The brain’s learning and adaptation processes heavily rely on the concept of associative memory. One of the most basic associative learning processes is classical conditioning. This work presents a memristive neural network-based associative memory system. The system can emulate Pavlovian conditioning principles including acquisition, extension, generalization, differentiation, and spontaneous recovery that have not been considered in most of the previous counterparts. The proposed circuit can emulate these principles thanks to the resistance-changing characteristics of the memristor. Generalization has been achieved by providing both unconditional and neutral stimuli to the network to reduce the memristance of the memristor. Differentiation has been attained by employing unconditional and conditional stimuli in a training scheme to obtain a certain memristance that causes the network to respond differently to both stimuli. A revival of an exterminated stimuli is also done by increasing the synaptic weight of the system. Compared to previous designs, the proposed memristive circuit can implement all the functions of conditional reflex. Our rigorous simulations demonstrated that the proposed memristive system can condition neutral stimuli, show generalization between similar stimuli, distinguish dissimilarities between the generalized stimuli, and recover faded stimuli.

## 1. Introduction

With increasing demands for brain-like computing, neuromorphic systems have engrossed the attention of scientists. As a result, artificial neural networks (ANNs) have seen rapid growth in hardware and software implementation [1,2]. ANNs have been used to mimic biological characteristics, brain functions, and means of processing information. A neural network has a comprehensive range of applications and growth potential in many fields. Synaptic plasticity is an influential part of the study of ANN, by which it can become adaptive. Synaptic plasticity can be implemented by using digital, analog, or a combination of both hardware settings, which spends a large amount of area budget [3]. Therefore, ANN is generally fetched by software as it is challenging to scale up the hardware to an acceptable biological size to implement intricate bionic functionality.

From the hardware point of view, various semiconductor devices such as ferroelectric transistors, floating-gate devices, non-silicon artificial semi-conductors, and memristors can serve as nonvolatile devices [4,5,6,7]. Among these candidates, memristors have emerged as promising devices due to their nonvolatility, compactness, power efficiency, agility, and integration capability with the complementary metal–oxide–semiconductor (CMOS) manufacturing process [8,9]. A memristor can hold multiple values by varying its resistance and emulating artificial synapses in brain-inspired computing. Memristors can be used in immense applications such as neural networks, logic gates, image processing, and machine learning [10,11,12]. Therefore, memristors have emerged as a tool for building synaptic structures at the circuit level of ANNs.

Many studies have been conducted to investigate the integration possibilities of memristors with neural networks to explore associative memory [13,14]. The ability to recall relationships between concepts, as well as un-connected facts, is referred to as associative memory. Understanding the learning behaviors of an organism is possible through associative learning. As a result, numerous studies have established the principles of biological associative memory for learning and assessment [15,16]. Classical conditioning is a straightforward type of associative learning in which the conditioned stimulus modifies the behavioral response. Many experiments and case studies have been conducted to develop classical conditioning theory. Among these experiments, the validation of Pavlov’s experiment is known as the basis for the majority of the current research on associative memory using memristor-based ANNs [17].

Pavlovian classical conditioning theory had a profound impact on the understanding of the learning and forgetting process. Associative memory is the most fundamental and widespread form of learning both for humans and animals [18]. The proposed associative memory circuit can realize the fundamental principle of the Pavlovian conditional reflex principle. From the circuit-level point of view, associative systems had previously been implemented at software and processing levels. However, they are significantly budget-consuming and complex. As a result, a brand-new class of circuits known as “neuromorphic circuits” was created. The main goal of this work is to investigate the implementation possibilities of Pavlovian conditional reflex at the memristive circuit level. This provides a pathway for systems with chip implementation for more complex associative memory systems based on our proposed method.

Pavlovian conditioning was primarily implemented using a microcontroller and a memristor-based synapse by Pershin et al. [19]. Since this work does not have a specific learning method, researchers afterwards worked on modifying associative memory circuits for learning [20]. To explore more of the learning functionality, a method based on implementing learning rules was proposed by Chen et al. [15]. However, they did not include the forgetting functionality. Therefore, the forgetting function was explored by Hu et al. [14]. Nevertheless, they worked on only one type of forgetting. Liu et al. formulated two kinds of forgetting functions [21]. However, the rest of the conditional reflex principle was not explored. In an attempt to investigate further, generalization and differentiation principles were proposed in a feed-forward neural network [22]. However, this work does not explore spontaneous recovery.

Previously, we have focused on the fabrication of a doped HfO2 memristor and its application in memristive image edge detection hardware [23]. This work aims to propose a memristive system that can include all possible Pavlovian conditional reflex principles. The proposed implementation method incorporates reflex principles including acquisition, extinction, generalization, differentiation, and spontaneous recovery in order to explore all principles of Pavlovian conditional reflex in a single system. In addition to simulating acquisition and extinction, the proposed model can also demonstrate generalization and differentiation principles. Moreover, it shows a process of recalling extinct conditional stimuli (CS).

Following this introductory section, the research background including Pavlov’s experiment and the developed memristor model is presented in Section 2. The proposed memristive system is provided in Section 3. Mathematical descriptions and a performance evaluation are carried out in Section 4. Finally, Section 5 highlights the main achievements.

## 2. Research Background

### 2.1. Pavlov’s Experiment

Psychological phenomena related to associative learning are collectively called classical conditioning. One of the first studies to show the essential features of classical conditioning was the well-known experiments of Pavlov with dogs [17]. The dogs salivated in the presence of food, at the sight of food, an empty bowl, and even at the sound of lab assistants’ footsteps. This “acquisition” happens when an organism learns to link a neutral stimulus and an unconditional stimulus (UCS) [24]. Once these external stimuli were presented repeatedly without food, the dogs faced extinction of learned behavior. Pavlov stimulated a dog by showing food from a distance but not giving it. The dog initially salivated, but the conditional response (CR) died ultimately. Nevertheless, the CR revived once the UCS was provided again. This response is regarded as “spontaneous recovery” [24,25,26]. Later, Pavlov worked on “generalization” and “differentiation” [24]. Pavlov worked with various tactile stimuli, such as scratching, rubbing, stroking, and pressing. He used these complex stimuli as CS and hydrochloric acid as UCS. After a series of experiments with different dogs, Pavlov found that when a dog had CS enforced in the form of scratching, it also provided a CR when pressure was applied in the same area. He described the event as a “generalization” due to the similar aspects of a stimulus with another one that stimulated the dogs [25]. Because of the complexity and shared components of all tactile sensory stimuli, these recurring components generalized the stimuli and enabled a novel stimulus to cause salivation. Even though the tactile stimuli were comparatively complex, they had specific characteristics and elements in each tactile stimulation. Therefore, with a repeated presentation, “differentiation” between distinct types of tactile impulses was possible. Pavlov also found that if tactile stimuli were repeatedly presented without being reinforced by an UCS, the CR gradually waned until it disappeared. Figure 1 shows an illustration of Pavlov’s experiment.

### 2.2. Memristor Model

A memristor is an electrical element that modulates and restricts the flow of electrical current in a circuit while also keeping track of the amount of charge that has passed through it. Popular memristor mathematical models include the linear ion drift model [27], the Simmons tunnel barrier model [28], the generalized model [29], the threshold adaptive memristor model [30], and the voltage threshold adaptive memristor model [31]. The above models do not describe the synaptic behavior of modern memristive devices in artificial neural circuit systems. In this work, we utilized a voltage-controlled threshold memristive model [32]. The mathematical memristor is defined as:(1)$$\frac{\mathrm{dw}\left(t\right)}{\mathrm{dt}}=\{\begin{array}{ll} u_{v}\frac{R_{\mathrm{ON}}}{D}\frac{i_{\mathrm{off}}}{i\left(t\right){-i}_{0}}f\left(w\left(t\right)\right), & v\left(t\right){>V}_{T+}>0\\ 0, & V_{T-}\leq v\left(t\right)\leq V_{T+}\\ u_{v}\frac{R_{\mathrm{ON}}}{D}\frac{i\left(t\right)}{i_{\mathrm{on}}}f\left(w\left(t\right)\right), & v\left(t\right){<V}_{T-}<0 \end{array}$$
where i_off,_ i_0_, and i_on_ are equation constants, V_T+_ and V_T−_ are threshold voltages of opposite polarity, and u_v_ is the migration rate of the impurity layer, respectively. R_ON_ is the low resistances of the memristor, D is the size, and w (t) is the doped region of the memristor.

The nonlinear ion drift phenomenon of the modeled memristor is expressed by [33]:(2)$$\;f\left(w\left(t\right)\right)=1-{\left(\frac{2w\left(t\right)}{D}-1\right)}^{2p}$$

Figure 2 shows the gradual resistance variation of the modeled memristor with respect to the input signal. The developed model has been implemented using Verilog-A in the Cadence Virtuoso environment. The positive and negative threshold voltages are V_T+_ = 5 V and V_T−_ = 1 V, respectively. According to Figure 2, when the positive terminal of the memristor is given a positive pulse greater than V_T+_, the memristance starts to decrease. Initially, the memristance decreases at a significant rate, and after that, it decreases at a slower rate. The opposite case happens when the negative terminal is given a negative voltage pulse less than V_T−_. The conductance is the reciprocal of memristance, and the change in conductance can accurately reflect the change in synaptic weight. Increasing (decreasing) the conductance of the memristor leads to the strengthening (weakening) of synaptic weight. Table 1 shows the fundamental parameters of the modeled memristor.

The proposed memristor model is based on the AgInSbTe, which was fabricated in [34], and the spice model was presented in [32]. The HP model [27] is unable to adequately represent the device behavior of memristors based on AgInSbTe. Therefore, the new spice model is used to obtain the characteristic of AgInSbTe-based memristors. The voltage applied at both ends affects the memristance of the model. The memristor’s resistance does not change when the applied voltage is withdrawn. Compared to previous memristor models, this model seems to be more appropriate for designing an artificial network circuit.

## 3. The Proposed Memristive System

The proposed system has a control circuit and four input and one output neurons. The input and output neurons are connected through two memristors and a resistor. The entire network is divided into three synapses (Synapse 1 to Synapse 3). Neuron one to Neuron four (N1–N4) are considered as food, scratching, spontaneous recovery, and stroking stimuli, respectively. The summation of N1 to N4 regulates the response of the output neuron. If the combination of the input stimuli and memristance (resistance) is greater than a certain threshold voltage, the output neuron produces an output. As food is UCS, a high weight should be connected to N1 to have an output when a productive input is present. As both scratching and stroking are learned behaviors [24], a variable weight should be connected, allowing for a weight change based on the given stimulus. As synapse weight determines the strength of the connection between two neurons, and food is a UCS, a resistor with low resistance (high conductance) is connected to N1 to have a significant connection weight. As scratching and stroking are both instances of CS and have to be learned [24], a memristor is connected to the rest of the input neurons. If CS is given along with a USC, the reaction appears, and if CS is given without UCS, the reaction eventually disappears. Therefore, the connection weight has to be variable. The scratching and stroking stimuli are learned and can be faded without food stimuli. As a result, memristors are employed to mimic the synaptic weight.

In order to feed the system with the input stimuli, a control circuit has been designed using complementary logic gates (Figure 3). As depicted, the first operating signal of the synaptic network (V_OP1_) has been established through the OR operation of food (N1) and scratching (N2) stimuli. A two- input AND gate (AND1) serves to prepare the associating signal (V_A_) related to the food and scratching stimuli. The AND2 logic gate collects the data from all input stimuli to produce the recovery signal (V_R_) of the synaptic network. The AND3, AND4, and OR2 logic gates have been exploited to generate the second operating signal (V_OP2_) related to the food, scratching, and stroking stimuli. The third operating signal (V_OP3_) is used to generate the related control signal when only food and scratching (not stroking) stimuli are in operation.

Table 2 shows the truth table for all combinations that have been used in the work. Initially, all the stimuli are given separately and the responses of all the gates are recorded. Afterwards, combinations of more than one stimulus are considered and the responses of the logic gates are noted. It is worth mentioning that (↑) and (↓) are considered 1 and 0, respectively.

Figure 4 represents the modeled Synapse 1 (related to the food neuron) using a threshold (1 V)-dependent switch and a resistor. When the input is fed with a 5 V magnitude signal, the SW1 switch turns on and provides 7 V to the output. The RF = 80 Ω is in a series connection with SW1 to translate the input stimuli to the synapse output.

Figure 5 portrays the modeled Synapse 2 (related to the scratching, spontaneous recovery, and stroking neurons) having two p-type (T1, T2) and two n-type (T3, T4) transistors. When operating signal 1 (V_OP1_) is in a high state, T1 and T3 are on, and T2 and T4 transistors enter the cut-off region. Correspondingly, reducing V_OP1_ to 0 V turns on T2 and T4 transistors and deactivates T1 and T3 transistors. 

The modeled Synapse 2 contains four switches (SW2 to SW5) with a threshold voltage of 1 V, six resistors (R1 to R6) with a 1 KΩ resistance, one memristor (M1), one summing operational amplifier (Amp1), and one inverting operational amplifier (Amp2). When the second neuron (N2) is in operation, the SW3 turns on and passes V3 = 4.5 V to the memristor navigating circuit. Since the V3 voltage (4.5 V) is smaller than the positive threshold voltage (VT_+_ = 5 V), memristance of the memristor is not changed. However, when the acquisition voltage signal, V_A_ (5 V), and N2 are in operation, the accumulated voltage is now greater than the positive threshold voltage of the memristor. Therefore, it forces the memristor to decrease the memristance and increase the connection weight between N2 and the output neuron. Similarly, when N3 is in operation, generates a recovery signal (V_R_), and feeds the input of the SW4 with V_R_, the SW4 turns on and passes V4 voltage (2 V) to the memristor navigating circuit. Since the V4 voltage (2 V) is smaller than the positive threshold voltage, memristance does not change once again. However, when N3 and N2 are in operation, the total voltage is greater than the positive threshold voltage. As a result, the memristance changes, and the connection weight between N2, N3, and the output neuron increases. N4 goes directly to the negative terminal of M1. When the V_OP1_ is low and N4 is in operation, V5 (2 V) goes to the negative terminal. In such a case, the memristance decreases when a voltage less than −5 V is given. On the contrary, the memristance increases once a voltage greater than −1 V is given. As a result, V5 is sufficient to change the memristance.

Figure 6 represents the modeled Synapse 3. The modeled Synapse 3 contains a switch (SW6) with a threshold voltage of 1 V, the Memristor 2 (M2), and two p-type (T5, T6) and two n-type transistors (T7, T8). When operating signal 3 (V_OP3_) is high, T5 and T7 are on, and T6 and T8 transistors enter the cut-off region. Correspondingly, reducing V_OP1_ to 0 V turns on T6 and T8 and deactivates T5 and T7 transistors. When N4 is involved and both operating signal 2 (V_OP2_) and V_OP3_ are in a high state, SW6 turns on and passes V6 (6.2 V) to the positive terminal of M2. When V_OP3_ is in a low state and V_OP2_ is in a high state, V6 goes to the negative terminal of the M2 memristor.

The circuit-level implementation of the output neuron is illustrated in Figure 7. It consists of one summing amplifier (Amp3), one inverting amplifier (Amp4), one comparator (Amp5), and four resistors (R7 to R10). The output of the summing amplifier, Amp3, is negative, and to invert the outcome of Amp3, an inverter, Amp4, is used. Amp5 is a comparator with a threshold voltage of V_REF_ (90 mV). 

When the output of Amp4 passes the threshold voltage of Amp5, the output signal of the comparator provides a high state (1 V), which means that the output neuron is activated. On the contrary, if the Amp4 output voltage is lower than the threshold voltage, the output neuron is not activated. This is similar to feed-forward propagation in neural networks.

In the following, the comprehensive mathematical description and performance evaluation of the proposed system are provided. It is worth mentioning that all the simulations have been performed in the Cadence Virtuoso environment.

## 4. Mathematical Description and Performance Evaluation

Figure 8 shows the complete schematic of the proposed circuit for Pavlovian conditional reflex. As depicted, the proposed design has five AND gates, two OR gates, and five NOT gates. All the voltage-controlled switches (SW1 to SW6) have a 1 V threshold voltage. The switches deliver V1 to V6 voltages to the synaptic network. Moreover, there are five op-amps to perform summation (Amp1 and Amp3), inversion (Amp2 and Amp4), and comparison (Amp5) operations. V_OP1_, V_OP2_, V_OP3_, V_R_, and V_A_ represent the operating signal 1, signal 2, signal 3, recovery, and associating signal, respectively. Memristor 1 (M1) and Memristor 2 (M2) have an initial value of 6.5 KΩ. The entire circuit contains an output neuron portion, three synapse portions, and a control circuit.

Table 3 provides the critical description of each simulation framework. It is worth noting that ”1” and ”0” binary digits represent activated and deactivated states of signals, respectively.

Initially, in Step 1, only the N1 is activated and turns SW1 on, which passes V1 to the input terminal of the output neuron. The input of the comparator (Amp5) is determined as:$$v_{\mathrm{Amp}3}=-\frac{R_{7}}{R_{F}}\times v_{1}=-2.625\;V$$
$$v_{\mathrm{Amp}4}=-\frac{R_{9}}{R_{8}}\times v_{\mathrm{Amp}3}=2.625\;V$$

As threshold voltage V_REF_ = 90 mV, the comparator is in operation and activates the output neuron.

In Step 2, only scratching (N2) is given. Therefore, the SW3 turns on and delivers V3 to Amp1. Moreover, the high state of V_OP1_ turns on T1 and T3 transistors. Consequently, the V3 is fed to the summing amplifier, Amp1. In this regard:$$v_{\mathrm{Amp}1}=-\frac{R_{4}}{R_{2}}\times v_{3}=-4.5\;V$$
$$v_{\mathrm{Amp}2}=-\frac{R_{6}}{R_{5}}\times v_{\mathrm{Amp}1}=4.5\;V$$
$$v_{\mathrm{Amp}3}=-\frac{R_{7}}{R_{M1}}\times v_{\mathrm{Amp}2}=-20.7\;\mathrm{mV}$$
$$v_{\mathrm{Amp}4}=-\frac{R_{9}}{R_{8}}\times v_{\mathrm{Amp}3}=20.7\;\mathrm{mV}$$

The output of V_Amp4_ is 20.7 mV, which is lower than V_REF_ = 90 mV of Amp5 and, as a consequence, the output neuron is not activated.

In Step 3, only spontaneous recovery (N3) is given, which is connected to a four-input AND gate, and it only works when N1 and N4 are off and N2 and N3 are on. As long as N2 stimulus is not in operation, nothing comes to Amp1, and the output neuron is not activated. 

In Step 4, only the stroking (N4) neuron is activated. Therefore, SW5 turns on and passes V5 (2 V) to the negative terminal of Memristor 1 (M1). The low state of V_OP1_ turns on T2 and T4 transistors. However, M1 is in the ROFF state (high-resistance state). Since applying a voltage greater than the threshold voltage for the negative terminal causes the memristor to increase the memristance, there is no scope for increasing the memristance beyond the high-resistance ROFF state. Furthermore, a high state of V_OP2_ activates SW6, and a low state of V_OP3_ turns on T6 and T8 transistors. However, the second memristor (M2) is also in the ROFF state, so its memristance retains the previous value. The output of V_Amp4_ in this step is insufficient to turn on Amp5 and, therefore, the output neuron is not activated.

Figure 9 shows these four steps where food (N1), scratching (N2), spontaneous recovery (N3), and stroking (N4) stimuli are in operation. Each stimulus is presented for 36 s, and the stimuli comprise six cycles each. When a food stimulus is given, the output neuron produces an output.

In Step 5, food (N1) and scratching (N2) stimuli are given. As V_OP1_ is in a high state, the T1 and T3 transistors turn on. In addition, the V_A_ signal turns on the SW2, which supplies V2 (5 V) to Amp1. The N2 turns on the SW3, which provides V3 (4.5 V) to Amp1. Moreover, the high state of V_OP2_ signal turns on the SW6. The high state of operating signal V_OP3_ turns on the T5 and T7 transistors, and the input of the comparator op-amp (Amp5) is determined as:$$v_{\mathrm{Amp}3}\;=-\left(\;\frac{R_{7}}{R_{F}}\times v_{1}\right)-\left(\frac{R_{7}}{R_{M1}}\times v_{\mathrm{Amp}2}\right)-\left(\frac{R_{7}}{R_{M2}}\times v_{6}\right)=-5.79\;V$$
$$v_{\mathrm{Amp}4}=-\frac{R_{9}}{R_{8}}\times v_{\mathrm{Amp}3}=5.79\;V$$

The output of V_Amp4_ is 5.79 V, which is higher than V_REF_ = 90 mV of Amp5. As a consequence, the output neuron is activated.

In Step 6, only scratching (N2) stimulus is given to the system. The high state of operating signal V_OP1_ turns on the T1 and T3 transistors. The SW3 provides V3 (4.5 V) to the system. Although the voltage V3 is less than the positive voltage threshold of M1, the unvarying memristance allows the output neuron to have an outcome. The input of the comparator op-amp (Amp5) is determined as:$$v_{\mathrm{Amp}3}=-\frac{R_{7}}{R_{M1}}\times v_{\mathrm{Amp}2}=-0.88\;V$$
$$v_{\mathrm{Amp}4}=-\frac{R_{9}}{R_{8}}\times v_{\mathrm{Amp}3}=0.88\;V$$
where V_Amp4_ = 0.88 V, which is higher than V_REF_ = 90 mV of Amp5, activating the output neuron.

In Step 7, only stroking (N4) is given. The low state of V_OP1_ turns on the T2 and T4 transistors. Further, the V_OP2_ activates the SW6, and a low V_OP3_ state turns on the T6 and T8 transistors. Since applying a voltage greater than the threshold voltage of the modeled memristor to the negative terminal increases the memristance, the input of the comparator op-amp (Amp5) with the latest memristance is determined as:$$v_{\mathrm{Amp}3}=-\left(\;\frac{R_{7}}{R_{M1}}\times v_{5}\right)-\left(\;\frac{R_{7}}{R_{M2}}\times {\;v}_{6}\right)\;=-0.105\;V$$
$$v_{\mathrm{Amp}4}=-\frac{R_{9}}{R_{8}}\times v_{\mathrm{Amp}3}=0.105\;V$$

The value of V_Amp4_ is 0.105 V, which is higher than V_REF_ = 90 mV of Amp5, and this leads to the firing of the output neuron.

From Step 5 to Step 7, the generalization process is demonstrated. In the original experiment, both food (N1) and scratching (N2) were given simultaneously to train dogs to associate scratching with food. After conditioning, the dog salivated even with only scratching stimuli. However, tactile stimuli are complex and difficult to differentiate; hence, stroking stimuli also produced salivation.

Figure 10 shows the transient response of Steps 5, 6, and 7. Step 5 represents the training period and Steps 6 and 7 represent the testing periods of scratching (N2) and stroking (N4) stimuli, respectively. Initially, food (N1) and scratching (N2) stimuli are given to train the system. These are given for 25 cycles and 150 s. The given voltage is more than the positive threshold of M1 and M2. It causes the memristance of the memristors to decrease and triggers the comparator to provide an output. As the memristance reduces, the connection strength between N2 and the output neuron increases. When only the N2 is presented for 36 s, the output neuron still provides the related response since the memristance does not change due to the insufficient voltage given by the N2 stimuli to the memristor. When only stroking (N3) is given for 36 s, the memristance of both memristors increases. However, the memristance of both memristors is not high enough to hamper the system from producing an output.

Step 8 comprises three rounds of trials, and each has two stages. Initially (Stage 1), food (N1) and scratching (N2) stimuli are given together, and in the latter stage (Stage 2), only stroking (N4) is given. The first stage acts similar to Step 5 and the input of the comparator op-amp (Amp5) with M2 is determined by
$$v_{\mathrm{Amp}3}=-\left(\;\frac{R_{7}}{R_{F}}\times {\;v}_{1}\right)-\left(\frac{R_{7}}{R_{M1}}\times v_{\mathrm{Amp}2}\right)-\left(\frac{R_{7}}{R_{M2}}\times v_{6}\right)=-3.4517\;V$$
$$v_{\mathrm{Amp}4}=-\frac{R_{9}}{R_{8}}\times {\;v}_{\mathrm{Amp}3}=3.4517\;V$$

The later stage behaves similar to Step 7 and can be obtained as:$$v_{\mathrm{Amp}3}=-\left(\;\frac{R_{7}}{R_{M1}}\times v_{5}\right)-\left(\;\frac{R_{7}}{R_{M2}}\times v_{6}\right)\;=-0.1019\;V$$
$$v_{\mathrm{Amp}4}=-\frac{R_{9}}{R_{8}}\times {\;v}_{\mathrm{Amp}3}=0.1019\;V$$

Each of the three rounds can be sorted out following Steps 5 and 7. In each round of Step 8, the comparator receives more than the threshold voltage as its input, so the output neuron produces an outcome in every case.

Steps 9 and 10 are both testing steps, and these steps show how the system could differentiate the tactile stimuli and render outputs according to their prospective characteristics. In Step 9, only stroking (N4) is given and the input of the comparator op-amp (Amp5) with the latest memristance is determined as:$$v_{\mathrm{Amp}3}=-\left(\;\frac{R_{7}}{R_{M1}}\times {\;v}_{5}\right)-\left(\;\frac{R_{7}}{R_{M2}}\times {\;v}_{6}\right)\;=-0.0746\;V$$
$$v_{\mathrm{Amp}4}=-\frac{R_{9}}{R_{8}}\times v_{\mathrm{Amp}3}=0.0746\;V$$

According to calculations, the output of V_Amp4_ is 74.6 mV, which is lower than the reference voltage of Amp5 (90 mV), and the output neuron does not fire.

In Step 10, only scratching (N2) is given and the input of the comparator op-amp (Amp5) with M2 is given by:$$v_{\mathrm{Amp}3}=-\frac{R_{7}}{R_{M1}}\times {\;v}_{\mathrm{Amp}2}=-0.092\;V$$
$$v_{\mathrm{Amp}4}=-\frac{R_{9}}{R_{8}}\times {\;v}_{\mathrm{Amp}3}=0.092\;V$$

The output of V_Amp4_ is 91.6 mV, which is higher than the reference voltage of Amp5. Therefore, the output neuron starts to engender the related response. 

Steps 8 to 10 can demonstrate the differentiation process. The researchers found through the Pavlovian reflex experiment that although tactile stimuli are difficult to differentiate, there are some specific characteristics within every stimulant. In the experiment, as the dogs were frequently exposed to the same stimuli, they eventually learned to differentiate the dissimilarities between stimuli. As a result, they eventually salivated only when scratching (N2) was given, which was initially associated with food (N1).

Figure 11 depicts the transient response of Steps 8, 9, and 10. Step 8 consists of three rounds, and each round consists of stages: (1) food and scratching are given, and (2) only stroking is given. Stages 1 and 2 are given for six and three cycles, respectively. The three rounds ran from 0 to 162 s. Stage 1 decreases the memristance of both memristors, which makes a strong connection between N2 and the output neuron. As connection strength increases, the system delivers an output at this stage. Stage 2 forces a rise in memristance for both memristors, which indicates a reduction in the connection strength of presynaptic neurons and postsynaptic neurons. However, the connection remains strong enough to produce an output in Stage 2 as well. Step 9 starts from 198 s and ends at 216 s and comes only with a stroking (N4) stimulus. As the connection weight decreases, the stroking stimulus does not provide an outcome when an interval of 36 s is introduced to the system to mimic the actual experiment of Pavlov. However, the memristance of M1, after several fluctuations, was low enough to provide an output when scratching (N2) is given alone for 36 s in Step 10.

Step 11 and onwards is a continuation of Step 10. Memristance of both memristors starts from the end of Step 10. Step 11 and Step 12 are testing phases to check the behavior of the system. In Step 11, only stroking (N4) is given, and it works in the same way Step 7 works. The input of the comparator op-amp (Amp5) with latest memristance is determined as:$$v_{\mathrm{Amp}3}=-\left(\;\frac{R_{7}}{R_{M1}}\times {\;v}_{5}\right)-\left(\;\frac{R_{7}}{R_{M2}}\times {\;v}_{6}\right)\;=-0.05738\;V$$
$$v_{\mathrm{Amp}4}=-\frac{R_{9}}{R_{8}}\times {\;v}_{\mathrm{Amp}3}=0.05738\;V$$

The Amp4 produces 57.3 mV, which is lower than 90 mV reference voltage of Amp5. In this regard, the output neuron does not fire.

In Step 12, only scratching (N2) is given and the input of the Amp5 is defined as:$$v_{\mathrm{Amp}3}=-\frac{R_{7}}{R_{M1}}\times {\;v}_{\mathrm{Amp}2}=-0.05661\;V$$
$$v_{\mathrm{Amp}4}=-\frac{R_{9}}{R_{8}}\times {\;v}_{\mathrm{Amp}3}=0.0566\;V$$

The output of Amp4 is 56.6 mV, and compared to the reference voltage of Amp5, the output neuron is not activated.

Step 13 consists of activating two stimuli: scratching (N2) and spontaneous recovery (N3). When both stimuli are given, the memristance decreases and allows the output neuron to produce the related response.
$$v_{\mathrm{Amp}3}=-\frac{R_{7}}{R_{M1}}\times {\;v}_{\mathrm{Amp}2}=-0.6325\;V$$
$$v_{\mathrm{Amp}4}=-\frac{R_{9}}{R_{8}}\times {\;v}_{\mathrm{Amp}3}=0.6325\;V$$

Based on the calculations, the output neuron starts to fire the related output.

Based on the description, it can be inferred that step 12 shows the extinction of a conditional response. Moreover, step 13 indicates the retrieval of the response after a period of time following the extinction. Furthermore, the spontaneous recovery principle is demonstrated using steps 11 to 13. 

According to the Pavlovian experiment, the dogs would salivate in response to a sudden ring (CS) of a bell even after enough time had passed since the last extinction of the CR. Once both spontaneous recovery and scratching stimuli are given, the conductance improves. As a result, the output neuron engenders the related response.

Figure 12 shows the transient response of Steps 11, 12, and 13. Initially, stroking (N4) is given in Step 11 for 30 s, and it increases the memristance of both memristors. As memristance increases, the connection strength of the neurons decreases. As a result, the stroking stimulus does not provide any output. Step 12 has a scratching (N2) stimulus given for 30 s; the output neuron is not activated since memristance is high. Step 13 is given after a period of time (60 s) to mimic the experiment, which reduces the memristance of M1 and eventually makes a strong connection between the N2, N3, and the output neuron. Therefore, the output neuron produces the related response.

Our demonstrations show that the system can associate neutral stimuli with UCS and make them CS. Initially, a neutral stimulus is given with UCS to invoke the association phase, which makes the novel stimulus a CS. The system ultimately forgets to produce an outcome when only CS is given. Shortly afterwards, following the extinction of the association, once CS is given again, the system renders an output. Because tactile stimuli are difficult to distinguish, they can elicit an outcome even when not conditioned. However, each stimulus has specific characteristics that can be discerned after several encounters. These propositions can be demonstrated using the proposed system and Steps 1 to 13 are consistent with the Pavlovian conditional reflex experiment.

## 5. Conclusions

The goal of this work was to build a system in a memristive neural network that can mimic the principles of Pavlovian conditional reflexes such as acquisition, extinction, generalization, differentiation, and spontaneous recovery. This exploration was inspired by experiments conducted by Pavlov and his assistants. Initially, it explains the connection strength of different neurons and the reasoning behind using resistors and memristors as synapses. It also describes how using a memristor has helped to achieve various connection strengths that assist in explaining the system behavior of weak and strong connections between neurons. Initially, the input neurons are initiated to show a preliminary response of the system. Afterwards, a training session is conducted to associate food and scratching and later, a test with scratching and stroking is performed. Both the association and testing phases invoke an outcome. However, after a while, once stroking is given, no salivation happens. Scratching, however, induces a reaction because the proposed system can distinguish between both tactile stimuli. Moreover, the system is capable of restoring the extinguished stimulus to demonstrate its recovery capabilities. Our simulations show varying synaptic weights matching real-world scenarios and help the system to be in agreement with the experiments conducted by Pavlov and his assistants.

## Figures and Tables

**Figure 1 micromachines-13-01744-f001:**
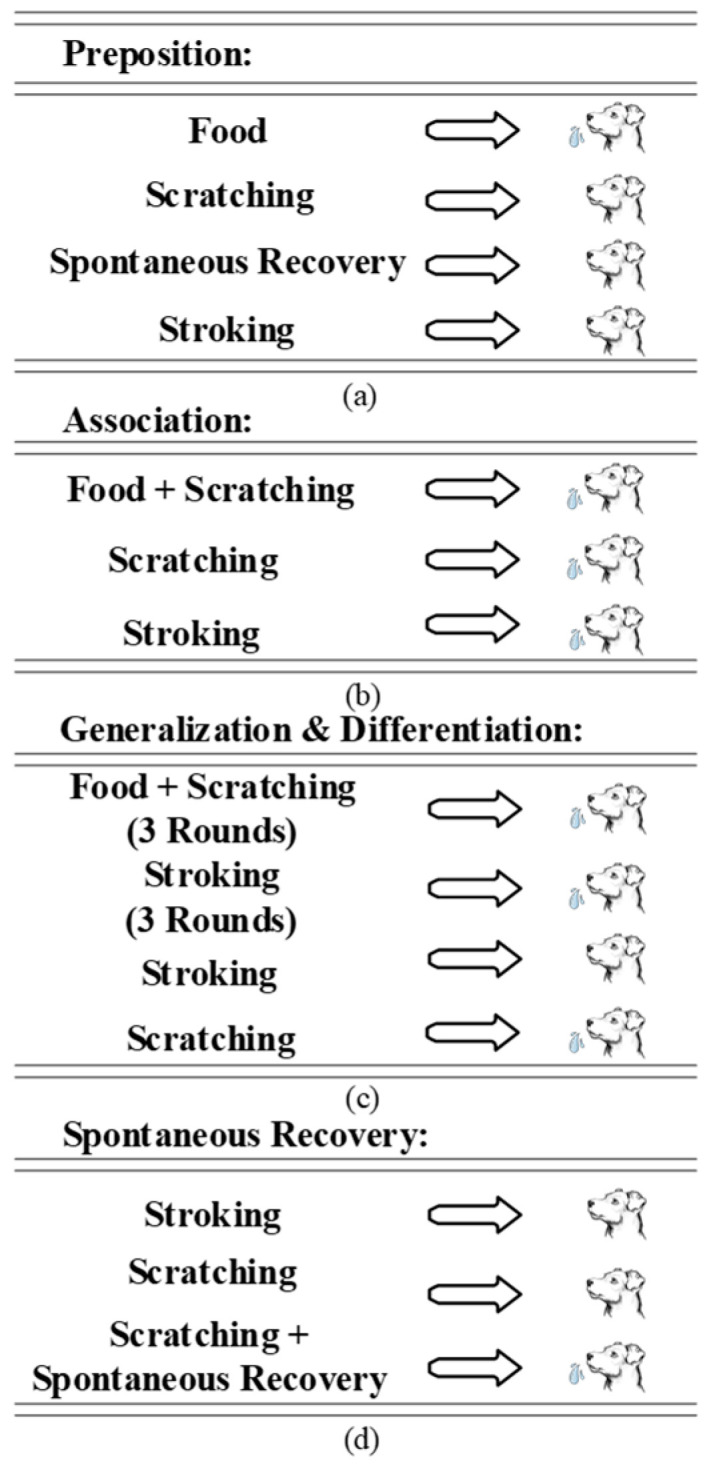
Illustration of Pavlovian conditional reflex. (**a**) Brief description of the preposition of the experiment. (**b**) Training scheme for dogs to associate unrelated things. (**c**) Generalization and differentiation experiment. (**d**) Spontaneous recovery principle.

**Figure 2 micromachines-13-01744-f002:**
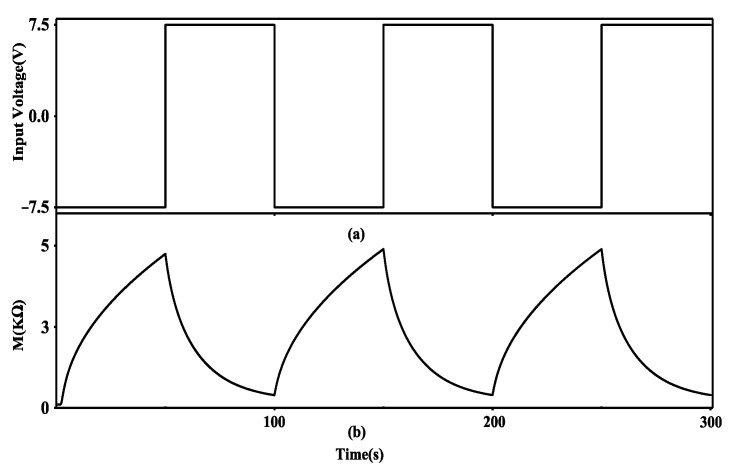
The memristance variation of the modeled memristor. (**a**) Input signal and (**b**) the memristance response.

**Figure 3 micromachines-13-01744-f003:**
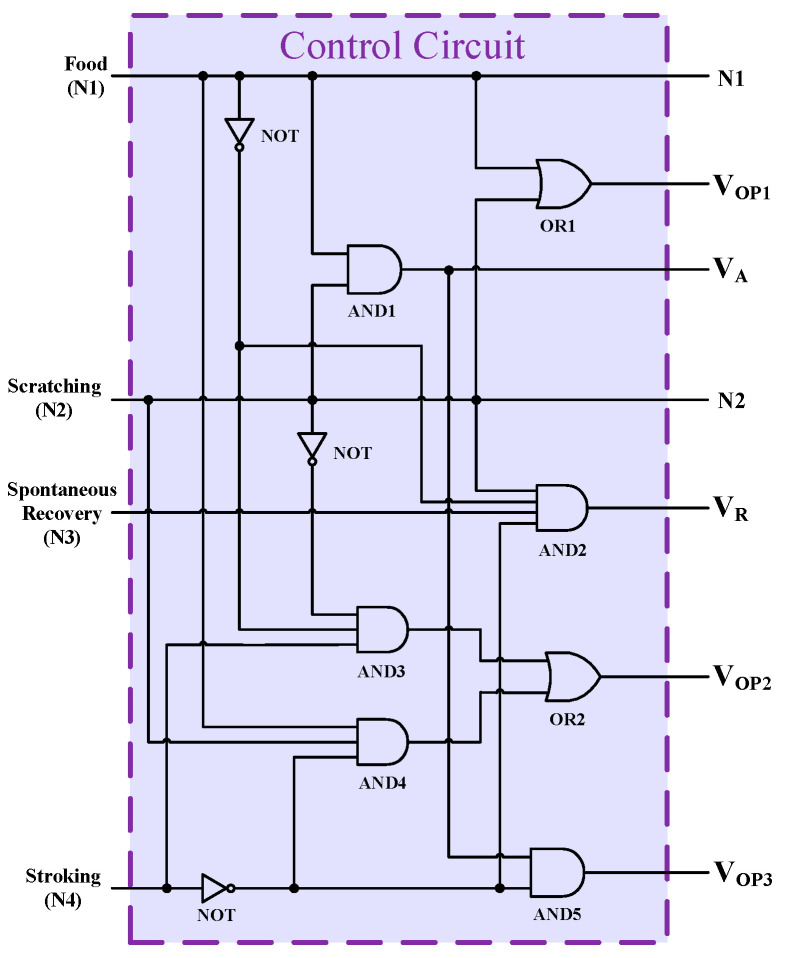
The schematic view of the proposed control circuit to feed synaptic network.

**Figure 4 micromachines-13-01744-f004:**
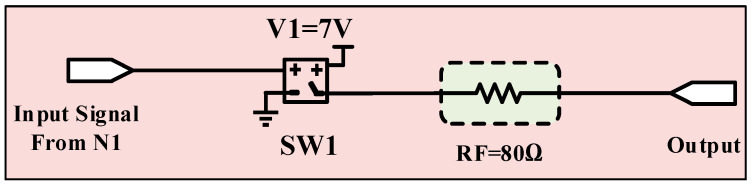
The first modeled synapse as a resistor and threshold-dependent switch.

**Figure 5 micromachines-13-01744-f005:**
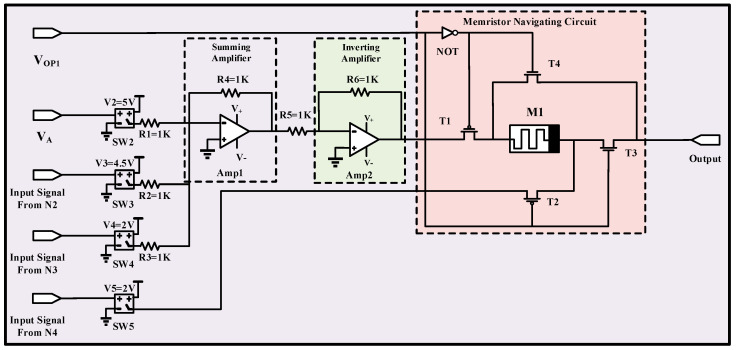
The circuit-level implementation of modeled Synapse 2 consisting of three input neurons (N2–N4).

**Figure 6 micromachines-13-01744-f006:**
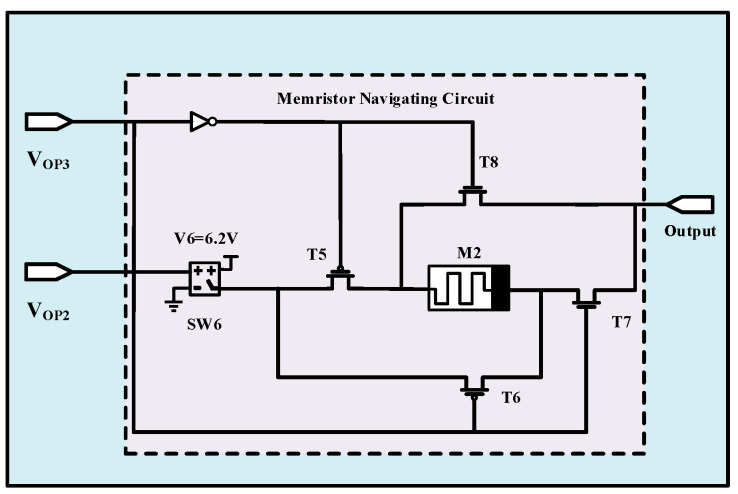
The circuit-level implementation of modeled Synapse 3 consisting of two operating signals, a switch, and Memristor 2 (M2).

**Figure 7 micromachines-13-01744-f007:**
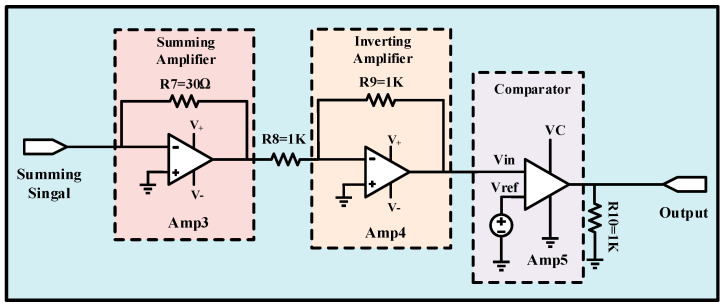
The circuit-level description of the output neuron.

**Figure 8 micromachines-13-01744-f008:**
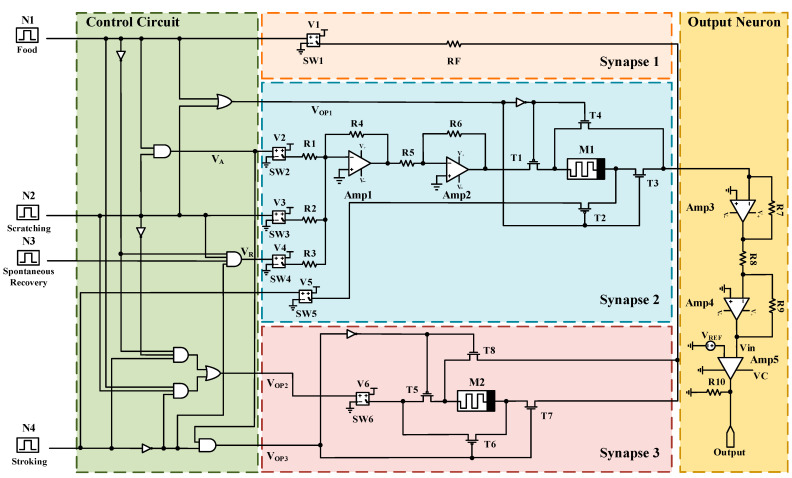
The schematic view of the proposed circuit for Pavlovian conditional reflex.

**Figure 9 micromachines-13-01744-f009:**
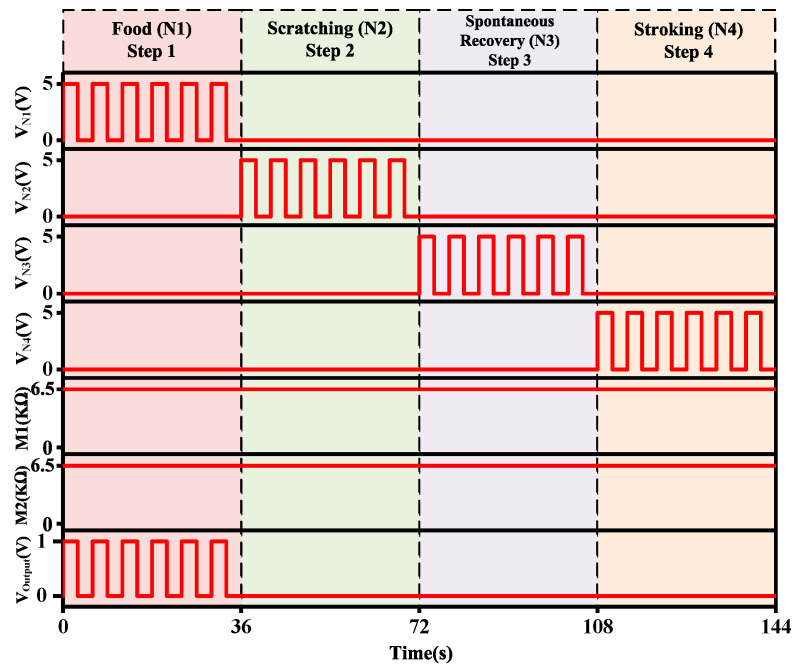
The initial transient response of each input neuron stimulus.

**Figure 10 micromachines-13-01744-f010:**
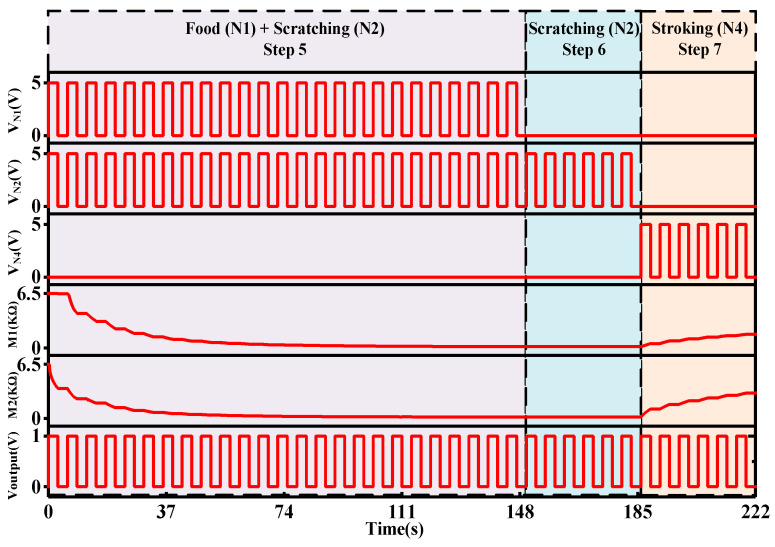
The transient response of Step 5 to Step 7.

**Figure 11 micromachines-13-01744-f011:**
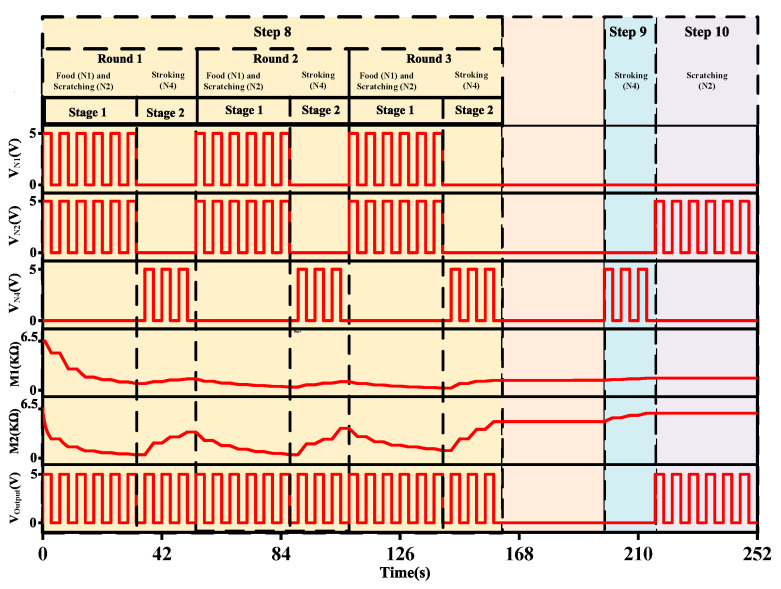
The transient response of Step 8 to Step 10.

**Figure 12 micromachines-13-01744-f012:**
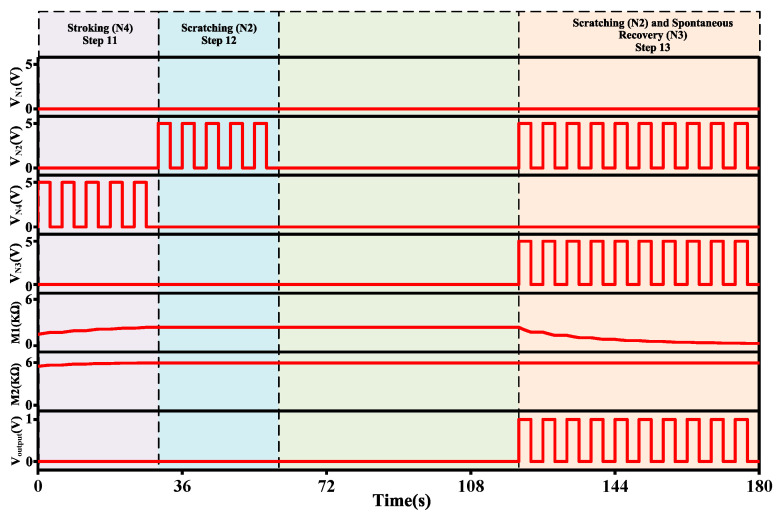
The transient response of Step 11 to Step 13.

**Table 1 micromachines-13-01744-t001:** The key parameters of the modeled memristor.

Parameters	Values
Initial current (i_0_)	1 mA
On current (i_on_)	1 A
Off current (i_off_)	10 μA
Memristor on-state resistance (R_ON_)	100 Ω
Memristor off-state resistance (R_OFF_)	6.5 KΩ
Positive threshold voltage (V_T+_)	5 V
Negative threshold voltage (V_T-_)	1 V
Physical width (D)	3 nm
Linear ion mobility (u_v_)	4.5 × 10^−19^ m

**Table 2 micromachines-13-01744-t002:** The truth table of the proposed control circuit.

	AND1	AND2	AND3	AND4	AND5	OR1	OR2
N1 (↑)	↓	↓	↓	↓	↓	↑	↓
N2 (↑)	↓	↓	↓	↓	↓	↑	↓
N3 (↑)	↓	↓	↓	↓	↓	↓	↓
N4 (↑)	↓	↓	↓	↓	↓	↓	↓
N1 (↑) + N2 (↑) + N4(↓)	↑	↓	↓	↑	↑	↑	↑
N1 (↓) + N2 (↓) + N4 (↑)	↓	↓	↑	↓	↓	↓	↑
N1 (↓) + N4(↓) + N2 (↑) + N3 (↑)	↓	↑	↓	↓	↓	↑	↓

**Table 3 micromachines-13-01744-t003:** The status of electrical components and signals during simulation.

Steps	V_N1_	V_N2_	V_N3_	V_N4_	V_OP1_	V_OP2_	V_OP3_	V_A_	M1(KΩ)	M2(KΩ)	V_Output_
Step 1	1	0	0	0	1	0	0	0	6.5	6.5	1
Step 2	0	1	0	0	1	0	0	0	Unvarying	Unvarying	0
Step 3	0	0	1	0	0	0	0	0	Unvarying	Unvarying	0
Step 4	0	0	0	1	0	1	0	0	Unvarying	Unvarying	0
Step 5	1	1	0	0	1	1	1	1	0.15	0.14	1
Step 6	0	1	0	0	1	0	0	0	Increasing	Increasing	1
Step 7	0	0	0	1	0	1	0	0	1.5	2.8	1
Step 8	1	1	0	0	1	1	1	1	Decreasing	Decreasing	1
	0	0	0	1	0	1	0	0	Increasing	Increasing	1
	1	1	0	0	1	1	1	1	Decreasing	Decreasing	1
	0	0	0	1	0	1	0	0	Increasing	Increasing	1
	1	1	0	0	1	1	1	1	Decreasing	Decreasing	1
	0	0	0	1	0	1	0	0	1.20	4.45	1
Step 9	0	0	0	1	0	1	0	0	1.47	5.28	0
Step 10	0	1	0	0	1	0	0	0	Unvarying	Unvarying	1
Step 11	0	0	0	1	0	1	0	0	2.38	5.77	0
Step 12	0	1	0	0	1	0	0	0	Unvarying	Unvarying	0
Step 13	0	1	1	0	1	0	0	0	0.31	Unvarying	1

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
