# Peer review of "Efficient Memristive Circuit Design of Neural Network-Based Associative Memory for Pavlovian Conditional Reflex"

_micromachines, 2022, doi:10.3390/mi13101744_

Round 1
Reviewer 1 Report
The authors have designed a memristor-based neural circuit and demonstrated its function of mimicking Pavlov’s experiments by simulations. The manuscript has a certain degree of scientific value. However, a major revision is needed before it can be accepted to publish in Micromachines. There are some comments:
1. The author should further highlight the significance of their research. Although the Pavlov’s experiments are important in the field of psychology, it seems not necessary to develop a customized electronic circuit that mimicking the responses of the dog.
2. The author used ideal device model in the simulation. However, there is a large gap between the model response and the electrical characteristics of the real device. So, the author should add a discussion about the model-data calibration in manuscript.
3. The author should add a truth table of the control circuit in Fig. 3
4. The workflow of the circuit is sophisticated and hard to follow. The author should add a waveform of some key points in the circuit (e.g. the resistance of the memristors, the output voltage of each synapse etc.). In current version, the authors only provide the waveform of Input & Output.
Minor Points:
1. Spelling & Typesetting errors. For example, Line 65 “for - getting”; Line 300 “How- ever”. Check it, please.
2. Line 101 “She used these complex stimuli as CS and hydrochloric acid as UCS.”
Line 103 “She described the event as a ”generalization” due to the similar aspects of a stimulus with another one that stimulated the dogs”
To my knowledge, Ivan Petrovich Pavlov is a male scientist. Check it, please.
Author Response
Dear Reviewer,
Thank you very much for your consideration and the time you spent to evaluate our manuscript. Especially, we highly appreciate the evaluation with the critical and reasonable comments. In response, we have corrected and revised the manuscript significantly and carefully. Now, we would like to resubmit our manuscript with the revised file which is attached below. For the convenience, we have merge the Author're reply letter and the Revised manuscript in Track Mode.
Thank you very much for your time and consideration one more time.
Yours sincerely,
Authors

Reviewer 2 Report
The manuscript designed a memristor circuit that can emulate the Pavlovian conditional reflex. The followings are some points to improve the quality of this manuscript.
1. In the caption of Figure 12, “This is a figure.” could be removed.
2. The manuscript lacks of power and area evaluations on the proposed circuits.
3. The simulated rising and falling edges of the waveform timing that the circuits produced seem too ideal. Did the authors consider any parasitic effects at a given CMOS technology node? What is the technology node the authors used in their circuit evaluation?
4. The background color of Figure 5 and Figure 7 seems too sharp.
Author Response

(The authors gave the same response as above.)

Round 2
Reviewer 1 Report
The author addressed all my issues, I recommend this article to publish on Micromachines. In view of there are seveal trival mistakes in the originonal manuscript, I think the author should check their mauscript carefully before publishing.
Reviewer 2 Report
The manuscript addressed all the questions.